# A Size, Weight, Power, and Cost-Efficient 32-Channel Time to Digital Converter Using a Novel Wave Union Method

**DOI:** 10.3390/s23146621

**Published:** 2023-07-23

**Authors:** Saleh M. Alshahry, Awwad H. Alshehry, Abdullah K. Alhazmi, Vamsy P. Chodavarapu

**Affiliations:** Department of Electrical and Computer Engineering, University of Dayton, 300 College Park, Dayton, OH 45469, USA; Alshahrys1@udayton.edu (S.M.A.); alshehrya1@udayton.edu (A.H.A.); alhazmia3@udayton.edu (A.K.A.)

**Keywords:** time to digital converter (TDC), field programmable gate array (FPGA), wave union, tapped delay line (TDL)

## Abstract

We present a Tapped Delay Line (TDL)-based Time to Digital Converter (TDC) using Wave Union type A (WU-A) architecture for applications that require high-precision time interval measurements with low size, weight, power, and cost (SWaP-C) requirements. The proposed TDC is implemented on a low-cost Field-Programmable Gate Array (FPGA), Artix-7, from Xilinx. Compared to prior works, our high-precision multi-channel TDC has the lowest SWaP-C requirements. We demonstrate an average time precision of less than 3 ps and a Root Mean Square resolution of about 1.81 ps. We propose a novel Wave Union type A architecture where only the first multiplexer is used to generate the wave union pulse train at the arrival of the start signal to minimize the required computational processing. In addition, an auto-calibration algorithm is proposed to help improve the TDC performance by improving the TDC Differential Non-Linearity and Integral Non-Linearity.

## 1. Introduction

Time-to-Digital Converters (TDCs) help to measure the precise time interval between two signals [1]. They are widely used in various applications including medical imaging (e.g., positron emission tomography (PET)) [2,3,4], High-Energy Physics (HEP) [3], Light Detection and Ranging (LiDAR) [5,6,7], robotics applications, self-driving vehicles [8], and time-of-flight (ToF) applications [6,9]. Further, in many applications, there is a need for high-performance, multi-channel TDC systems with short measurement dead time [10]. For low-channel count TDCs, Application-Specific Integrated Circuit (ASIC)-based TDCs have commonly performed better than advanced Field Programmable Gate Array (FPGA)-based TDCs, as ASICs can be low cost, small, and fully customized [11,12]. However, for high channel count TDCs, FPGA-based TDCs offer advantages of reconfiguration and short development time [13].

Typically, a TDC consists of a time reference generator which creates a reference signal with a fixed frequency, which is used as a timebase for the TDC. The TDC measures the time interval between the reference and input signals and converts the time interval into a digital output [1]. The measurement of the time delay between the reference signal and the input signal can be accomplished through various techniques, as listed in Table 1, which include Tapped Delay Line (TDL), coarse counters, pulse shrinking, and phased clocks [14]. Coarse counter-based TDC architecture requires a high-frequency clock to achieve high resolution [1]. Pulse shrinking-based TDCs are more suited to ASIC-based implementation [15]. Phased clock-based TDC architecture is susceptible to jitter, which limits the ability to reach high time measurement resolution [14,16]. TDL-based TDCs have become increasingly prevalent in FPGA-based implementation due to their efficient use of logic resources and provide high time resolution. This architecture utilizes the FPGA logic cells to introduce propagation delays. Multiple TDLs can be used by incorporating delay lines that are equivalent to the sampling and averaging the results [10,17,18,19]. To achieve high resolution, one must address the bubble error that arises with the TDL architecture. Bubble errors occur around the transition edges and cause uneven propagation in the FPGA logic structure [20].

Many techniques have been previously proposed solving bubbles errors in TDL TDCs for FPGA implementation [17,18,19,20]. Here, Wave Union (WU) approaches have been preferred due to their efficiency in enhancing the resolution and accuracy while reducing bubble errors [21]. WU launchers typically generate multiple logic transitions based on an input logic step and can be classified into Finite Step Response (FSR) (or type A) and Infinite Step Response (ISR) (type B) [22]. Again, the WU launcher type A (WU-A) is preferred due to its less complex decoding network, shorter dead time, simple calibration, and less timing jitter [23]. Thus, we chose Type WU-A as the launcher for our implementation, which consist of carry chain multiplexers and a Look-up Table (LUT).

Bubble errors in FPGA-based TDL TDCs occur due to metastability in flip-flops that are used to capture and store the timing information. It causes flip-flops to oscillate between 0 and 1 for a short time before settling to a stable value, which results in incorrect time measurements. Furthermore, the newer FPGA with large hardware resources, when used to implement TDCs, generates large differential non-linearity (DNL) and Integral Non-linearity (INL) due to the Logic Array Block (LAB) structure [24], firmware issues, or delays in the clock distribution network. Thus, overcoming the above problems typically requires calibration using a LUT. Additionally, the sensitivity of these parameters to Process, Voltage, and Temperature (PVT) variations may require regular updating of the LUT. The PVT fluctuations create uneven bin widths in TDCs, which increases the DNL [25]. However, the LUT implementation for the WU-A is a time-consuming process due to precise logic cell placing and routing [4].

**Table 1 sensors-23-06621-t001:** Overview of state-of-the-art TDCs.

RMS Resolution [Ps]	Channel	System	DNL (LSB)	INL (LSB)	Architecture	Ref.
2.9	16-Ch	Xilinx Spartan-6	N/A	19.36	Time Coding Line TCL	[26]
3.9	2-Ch Dual-Sampling	Xilinx Ultra-Scale	N/A	N/A	TDL	[19]
4.2	2-Ch (M = 8) Multi-Chain	Xilinx Virtex-6	3.8	19.36	Plain Tapped-Delay	[27]
4.5	8 -Ch	Xilinx Kintex-7	4.11	18.85	Time interval Counter	[28]
6	1-Ch	Custom ASIC	N/A	N/A	MASH TDC	[7]
7.4	1-Ch	Xilinx Virtex-5	1.4	3.09	Matrix of Counter	[29]
10	128-Ch	Xilinx Kintex-7	42	N/A	TDL	[10]
10.23	256-Ch	Xilinx Kintex-7	N/A	N/A	TDL	[18]
20	264-Ch	Lattice ECP3-150 EA FPGA	N/A	N/A	Delay Line TDC	[17]
69	64-Ch	ProASIC3	N/A	N/A	Coarse-Time Counters	[30]
81.3	4-Ch	Xilinx XC3S200AN	N/A	1.93	Pulse-Shrinking	[20]
198	8-Ch	Xilinx kintex-7	1.1	2.7	Multi-Phase Clocks	[16]

From Table 2, we notice that prior implementations of WUs on FPGA devices provided high-precision timing measurement. For example, Lusardi et al. [31] described a Super Wave Union 16-channel TDC implemented on the Artix-7 FPGA device with a time resolution of 12.5 ps. In [32], Kwiatkowski et al. presented Multi Sampling Wave Union to overcome the bubble error. Bayer et al. [20] applied a 48-channel based on WU-A, improving resolution with using a low supply voltage. Wang et al. [33] presented WU-A using carry chain multiplexers merged with TDL on Xilinx Kintex-7 and showed that a multi-edge encoding scheme could improve TDC performance. Liu et al. [10] designed a WU-A based on flip-flops using a single Kintex-7 device and achieved less than 10 ps Root Mean Square (RMS) time measurement resolution.

Our proposed method uses a Xilinx Artix-7 FPGA to improve size, weight, power, and cost (SWaP-C) efficiency for TDCs (See Table 3) [34,35]. Furthermore, our algorithm is designed to efficiently fit within the layout of eight clock regions (X0Y0 X1Y3) for Artix-7 FPGA, which offers better SWaP-C parameters than the more advanced FPGAs [33]. Our proposed solution utilizes the initial element (MUX 0) of WU-A to initiate the launcher. It connects the output of the initial element to the elements where the WU-A pattern exhibits a transition value from 0 to 1 and 1 to 0. This technique causes a time delay of one delay element and enhances the measurement precision. We notice that the proposed WU-A approach generates fewer spanned bins and, consequently, enhances the accuracy and the histogram of DNL outputs. This paper is organized as follows. Section 2 explains the architecture of the proposed WU-A launcher, and Section 3 describes the calibration algorithms. Section 4 and Section 5 provide implementation details and results, respectively. Finally, Section 6 concludes this work.

## 2. Proposed WU-A Launcher

Figure 1a [32,36] illustrates the commonly used WU-A architectures. From Figure 1a, the branch i design depends on carry multiplexers with TDL as shown by Wang, Y et al. in [19,37]. The branch ii design describes a WU-A based on using a LUT [4,31]. The branch iii design describes a WA-U built using flip-flops [10]. The data input of the carry multiplexer is set to a predefined value that specifies the WU pattern as a sequence of logical values.

Figure 1b shows our proposed WU-A architecture, which is used to generate wave union patterns. The proposed WU-A launcher is based on two CARRY4 blocks, as shown in Figure 2. Each CARRY4 logic cell path has four bits per slice [34], and each bit in these is a carry multiplexer (MUXCY cell in Artix-7). Here, the output of the first CARRY4 is connected to the next CARRY4. This pattern has a fixed logic transition. At each Start pulse, the logic transitions will propagate into the carry chain and record the data in the register arrays (flip-flops) for the primary encoding. With this method, the WU-A launcher can produce the first state of TDL output. At each edge of these transitions, the carry logic will have different snapshots of WU that is generated by the propagation through TDL carry chains [32].

In Xilinx FPGA devices [34,35], the logic fabric is split into Configuration Logical Blocks (CLBs). In a Series-7 (Aritx-7) FPGA device, each CLB has two slices that share a common switch matrix. Each slice has CARRY4, which contains four multiplexers, four LUTs with six inputs, and two outputs for eight flip-flops, as illustrated in Figure 2iii. In the proposed architecture, the four internal multiplexers are utilized as delay elements from (MUX_0 to MUX_3). The start signal is only connected to the first element (MUX_0), and the output of the first element is connected to the elements where the WU-A pattern has a transition value from 0 to 1 and 1 to 0, as shown in Figure 2ii and Figure 3iii. The advantage of this method is that the start signal will not arrive simultaneously at four delay elements, as shown in Figure 3. As we are working to develop a 32-channel TDC design, the output of the first delay element is also connected to the next TDC channel and so on from Ch01 to Ch32, as shown in Figure 3i,ii. This approach would require additional calibration steps to achieve high precision, as will be discussed in the following section.

## 3. Calibration

### 3.1. Calibration Algorithm

Using WU in conjunction with TDC necessitates the implementation of an appropriate decoding algorithm to interpret the TDL outputs. Specifically, the conversion outcome is encoded in a non-thermometric code, rendering the utilization of a basic thermometer-to-binary encoder inadequate for interpreting the transition from the TDL outputs. Given that the hit transition has been stored in the histogram, the total number of bins and the period of the clock reference can be determined. This approach would build a LUT that converts the bin number to calibrated time in picoseconds, assuming the clock period is 5000 ps (200 MHz). Then, the total bin number is 8192, binned in the histogram with hits represented by the width of the bin [$N_{i}=W_{i}$]. For example, (5000 ps/8192) = ($N_{i}$ * 0.6103 ps) for the 200 MHz system clock, as shown in Figure 4. Once all the information for the bins is present in the histogram, a sequence of controller cases builds the LUT [13]. Let us suppose that *W*${}_{k}$ is an array in which the TDC bin is measured and stored in Block-RAM memory [10,20]. The equivalent time value for each bin can be obtained from Equation (Equation 1) as:(1)$$Ti=\sum_{k=1}^{i-1}W_{k}+\frac{W_{i}}{2}$$

In our study, we utilize a multiplication process that involves the sampling value in the histogram and the value 65,536, which is equivalent to 2${}^{16}$ or 16 bits. The resulting product of this multiplication process is a value of 40,000. This value allows us to convert the bin number into a time value. However, during this process, there may be a delay in some values due to pipeline architecture [38]. After setting up the system, we calibrate all TDC bin numbers to record the DNL histogram and create the calibration LUT.

The calibration process is controlled by the same states. When a signal is received, the first step is to flush the histogram memory by storing 0 in every memory cell. Once the last cell in the histogram memory is set to 0, the process of booking the histogram begins. The second step involves booking 2${}^{N}$ samples (which is equivalent to 2${}^{16}$ in our design) in a histogram and counting how many times each bin number appears. The third step starts when the last sample is received and involves building up and integrating the LUT, as shown in Figure 4. Our objective is to enhance precision, so we delay each bin by multiplying its histogram value with the time constant and dividing it in half by the center of the bin number. To reduce measurement errors, we maintain the bin width when it varies and calibrate it to the center values. The final step is to build the LUTs in the FPGA internal memory and accumulate the memory data by the bin numbers during the LUT update.

### 3.2. Encoder

The TDL-TDC architecture is composed of two main parts, the TDL component, which is made with carry chains, which in Xilinx Artix-7 is (CARRY4), and the second component is the logic register, D Flip-Flop (DFF), as shown in Figure 5a. The bubble errors typically occur around the transition edge through the TDL. The leading cause of such errors is uneven propagation in the FPGA logic structure and AND gates. In particular, the bubble errors are recorded as a missing logic state (raising 1 or falling 0) near the transition. The bubble errors in the thermometer time code must be corrected [21,39]. Therefore, the output of the TDL needs to be converted from thermometer code to binary encoded (T2B).

In our design, a non-thermometer code to one-out-of-N (NTON) code converter was implemented using VHDL on the FPGA. The NTON employs a five-input AND gate with four inverted inputs followed by a DFF, as illustrated in Figure 5b. The second step of the encoder involves implementing the one-out-of-N code to binary code converter (ONBC) [39]. For instance, decoding [i] = [00100000…0] (i = 1, 2,⋯,2${}^{b}$) with a length of *b*. The algorithm for ONBC is represented by a block diagram, as shown in Figure 5c. Prior to detecting any decoded value [i][i] = 1, the input to the binary adder is ‘01’. Then, after detecting the decoded value [i][i] = 1, the input to the binary adder is ‘00’. Finally, the output ‘binary (2${}^{b}$)’ is the sum of all the binary-adder inputs, which represents the expected ‘edge’s number’. The register arrays (decoded value [i]) are synchronized using CLK signal. To reduce complexity while maintaining high speed, only the 32nd register array is clock-synchronized while the others are asynchronous. This process is repeatedly applied for all the stage registers until the transition bit is determined. The encoder output is a 16-bit binary number.

## 4. Implementation

### Layout of Single Channel and 32-Channel WU-A TDC

The 32-channel TDL-based TDC using the WU-A architecture is shown in Figure 6ii. Given that we are working to develop a timing measurement system with picosecond resolution, even the minute variations in the physical distances between any two-carry logic (CARRY4) blocks in the WU-A architecture will lead to high non-linearity. Therefore, a bin-by-bin calibration is necessary [40]. This part of the calibration using data processing is performed with an automatic calibration method. Then, the analysis is conducted using the code density method-based averaging algorithm, which is implemented using MATLAB [10,13]. The Artix-7 FPGA board (model: xc7a100tcsg324-1) used in this work is shown in Figure 6i [41].

The “floor planning” tool utilizes the necessary logic resources to implement the TDC system. The picture of the partial floor planning for both the single and 32-channel TDC using the WU-A method is shown in Figure 6ii,iii respectively. The 32-Channel TDC is constructed with all the available clock regions using the Vivado IDE software (version 2020.2). The green lines indicate the routing between the internal resources to build the TDL-TDC design. Table 4 shows on-chip resource utilization statistics to implement the proposed 32-channel TDL-TDC design. The 32-channel TDL-TDC employed up to 70% for resources on-chip. As shown in Table 4, LUT use was 73.86%, LUT RAM (LUTRAM) use was 32.05%, flip-flop use was 33.41%, and the block RAM (BRAM) use was 84.81% of the available BRAM to store the DNL histogram for the calibration algorithm.

## 5. Results and Discussion

The Artix-7 has a built-in 100MHz crystal oscillator and using the clocking wizard IP core, the reference clock (STOP) is set to 189.778 MHz. The START signal is set to 9.977 MHz to generate random time shifts between the two signals for measurements. Both signals are generated by a mixed-mode clock manager (MMCM) adjusted within the Vivado software.

### 5.1. Algorithm

The number N in Equation (Equation 2) represents the TDC output data, the least significant bit. The WU-A launcher has two transitions (0 to 1) which are the falling edge and (1 to 0) rising edge. With each transition, a high transition which generates 0–255 bits, and low transition which generates 256–512 bits, we obtain the total range (0 to 512-bits) TDL output [1].
(2)$$N=\frac{\Delta T}{T_{LSB}}$$
where Δ*T* is the time interval between start and stop events. By assuming the time delay of all the WU-A TDC bins is distributed evenly within one clock period, we can define Δ*T* as the time interval between start and stop events by Equation (Equation 3) [1].
(3)$$\Delta T=N*T_{LSB}+\varepsilon$$

Equation (Equation 4) calculates the delay time of a single delay element, $T_{LSB}$, which is the time of the least significant bit. The value of the LSB of the TDC is determined by the averaged bin size [27].
(4)$$T_{LSB}=\frac{\Delta T}{N}+\varepsilon$$
where ε is the quantization error that arises from reflecting the incorrect status of the flip-flops, which causes bubble error in the TDC outputs.

### 5.2. Time Measurements

We use the Tool Command Language script to measure the interval between two single START and STOP signals. The START signal is the reference clock to initiate the TDC system and the STOP signal is used to sample the TDC results. Accumulating the TDC results yields a histogram. After calibration, all the timing events are stored in the internal RAM of the Artix-7 FPGA. The total registered events are (2${}^{16}$ = 65,535), which are transferred to the computer through the JTAG interface for processing.

### 5.3. Average Process

For each TDC channel, the results are based on measuring the time interval between two rising edges. The output data contains the number of bits collected, which determines the time interval between two rising edges given to the TDC [20]. All the activated bin numbers are stored in the BRAM and then transferred to the computer to be processed using the code density method [10]. Since some bins have more counts than others, we use an averaging process by extracting the bin numbers from the raw output data using Equations (5) and (6). Due to the non-uniformed bin width of the delay times caused by PVT variations, we use the averaging process to find the least significant bit values of the measured time interval [27].
(5)$$T_{LSB}=\frac{T_{C}}{N}+\varepsilon$$
(6)$$T_{avg}=\frac{\sum_{i=n}^{N}avg\left(n\right)}{N}*T_{LSB}+\varepsilon$$
(7)$$RMS=\frac{T_{avg}}{\sqrt{2}}$$

The flowcharts in Figure 7 illustrate the sequence of signal processing steps applied to the TDC data to determine the average precision for a single delay element (CARRY4). We estimate the delay time for the single delay element within one clock period of the Sampling Frequency, $T_{C}$, using Equation (Equation 5). The averaged bin width is applied using Equation (Equation 6) to find the average number from all activated bins throughout the measurement events.

From Equation (Equation 6), the WU-A launcher introduces two transitions: a falling edge and a rising edge. The collected bin numbers are sorted from the smallest bin number to find the bins that successfully registered the STOP events. We calculate the average bin width for the least significant bit of the sampled data to build the histogram with a maximum 65,536 value.

### 5.4. Statistical Results

Table 5 lists the measurement results. This study computes the performance of four different scenarios, the average precision, worst case, and the best case, and utilizes Equation (Equation 7) to determine the RMS resolution of the measured time, which is obtained as 1.81 ps.

Figure 8 depicts the performance of the proposed TDC design. The inset located at the top-left quadrant displays the correlation between the number of activated bins in the carry chain and the hit counts.

Figure 9 shows a calibration curve that correlates the interval number with the delay bin term, corresponding to the total bins over time. The bin number can be converted into time intervals using the FPGA.

Figure 10 illustrates the absolute time intervals matching the elements of delay bins with the number of sampling and their distribution of timing intervals. The average value of each interval is 2.57 ps.

The histogram bars in Figure 11 show the distribution of the bins over the bin width in ps. From the figure, the total effective bins are 7827, which we use to inset one system clock period. The average bin width can be calculated by Equation (Equation 8), using the 1 LSB = 5.269 ns/7827 = 0.673 ps, as shown by Wang, Y et al. in [37]. Most TDC bins have a width around the average value, as shown in Figure 11.
(8)$$\mathrm{Average}\;\mathrm{Bin}\;\mathrm{width}\;\mathrm{at}\;1\mathrm{LSB}=\frac{\mathrm{Clock}\;\mathrm{Period}}{\mathrm{Effective}\;\mathrm{Bins}}$$

### 5.5. Test of Differential and Integral Non-Linearity

The measurement distribution of the TDC bin in terms of DNL and INL are visualized in Figure 12. The DNL and INL values are calculated by Equations (9) and (10) as follows [1,37,42]:(9)$$DNL\left[i\right]=\frac{W\left[i\right]-W_{LSB}}{W_{LSB}}$$
(10)$$INL\left[i\right]=\sum_{n=o}^{i-1}DNL\left[n\right]$$

The W[*i*] is the width of the measured i-th bin, and the $W_{LSB}$ is the LSB size. In our case, we measured 1 LSB = 673 fs. The DNL is calculated as the actual bin width minus the standard LSB size. The INL is defined as the summation of DNL for all active bins. As shown in Figure 12a,b, the differential non-linearity (DNL) for the described TDC is within [−1.01, +0.923] LSB. The DNL peak-to-peak shows an enhancement to 1.923 LSB. The INL in the range of [−25.44, +26.24] LSB, and the INL peak-to-peak of our TDC is in range of 51.68 LSB. A calibration table based on the measured INL is created to remove measurement errors [42].

### 5.6. On-Chip Power

Previously, Wu et al. [43] showed that the power supply noise negatively impacts TDC time measurement resolution. The power nets on our FPGA board are linear and allow better accuracy of time measurement. The power analysis from our 32-channel TDC is available in the power netlist. In the typical operating environment, the power supply voltage is 1.25 V on the FPGA core and is around 0.998V at VCCBRAM. The TDC activity caused at least 33.8 °C (on-chip) junction temperature and utilized 1.935 W total on-chip power in which dynamic consumption was 95% of power and static was 5% of the power consumption shown in Figure 13a,b.

## 6. Conclusions

We described the implementation of a multi-channel TDC using a SWaP-C efficient FPGA and demonstrated high-precision timing measurement. It includes a novel WU-A method by connecting only the first multiplexer as the input, and the output of this multiplexer is connected to the next multiplexers cascaded repeatedly to build the wave union launcher. Compared with other WU-A TDCs, our proposed method has some evident advantages, for instance, fewer spanned bins while improving the accuracy and the histogram of DNL outputs. Moreover, the presented Artix-7 FPGA utilized in our method has the advantage of small size and weight while maintaining low power consumption of 1.935 W. As a result, it achieved high precision by using an efficient WU-A algorithm and calibration method. The results show high accuracy with an average time measurement precision of 2.57 ps and (RMS) precision of 1.81 ps. The TDC (DNL) and (INL) were achieved within the range of [−1.01, +0.923] LSB and [−25.44, +26.24], respectively. The proposed TDC design demonstrates a suitable alternative for applications that require high precision time interval measurements with low size, weight, power, and cost (SWaP-C) requirements.

## Figures and Tables

**Figure 1 sensors-23-06621-f001:**
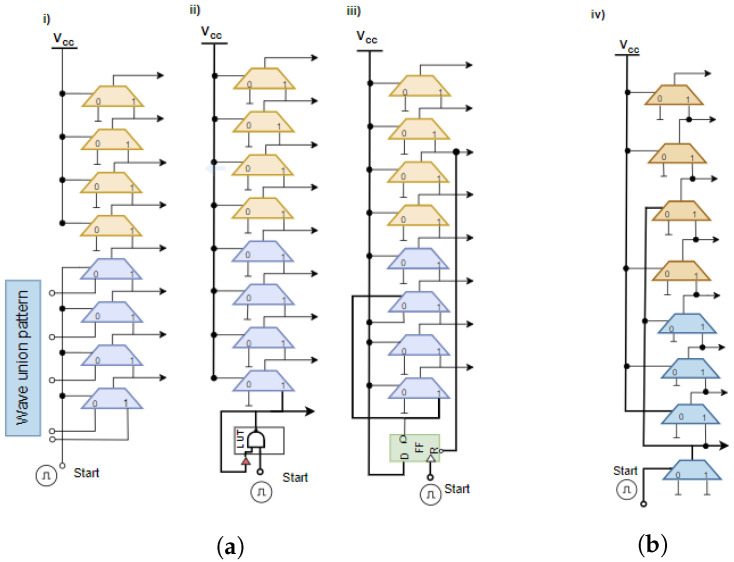
Comparison of proposed and standard WU-A architecture. (**a**) Standard WU-A architectures (**i**,**ii**,**iii**). (**b**) Proposed WU-A architecture (**iv**).

**Figure 2 sensors-23-06621-f002:**
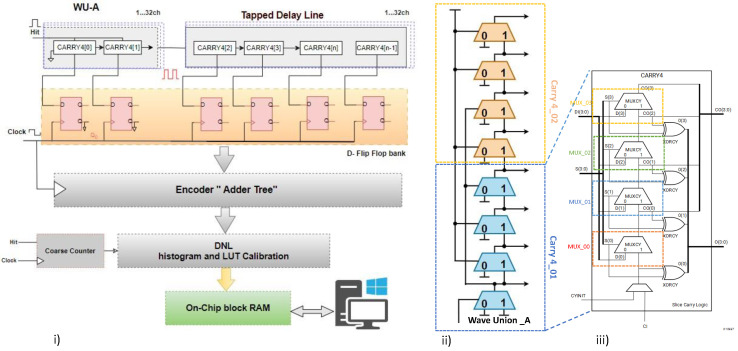
(**i**) Schematic diagram of complete TDC structure. (**ii**) Proposed WU-A architecture. (**iii**) Diagram of CARRY4 block of Xilinx Artix-7 FPGA.

**Figure 3 sensors-23-06621-f003:**
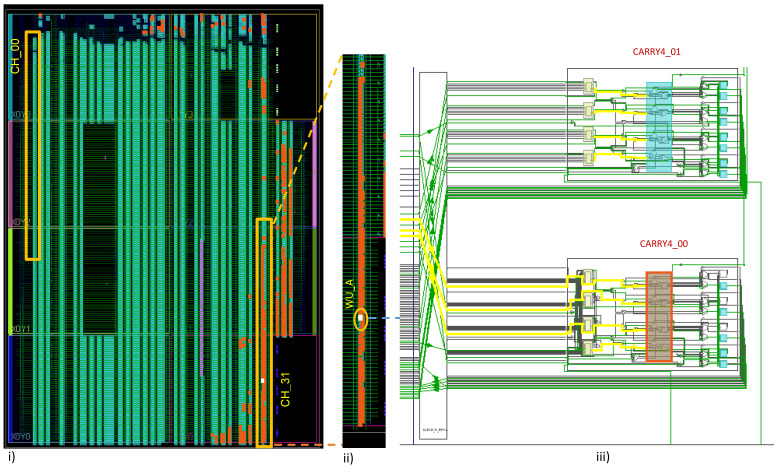
Implemention layouts of the WU-A TDL TDC. (**i**) Overview. (**ii**) Clock regions (X1Y0 X1Y1) containing TDC CH00. (**iii**) A single CLB for WU-A TDL TDC implementation.

**Figure 4 sensors-23-06621-f004:**
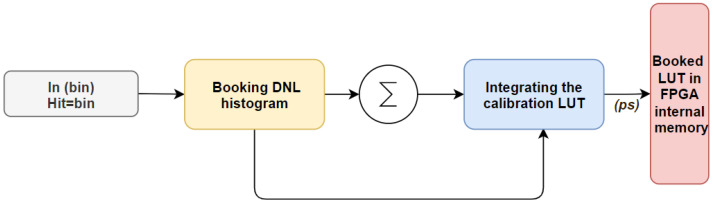
Automatic calibration functional block.

**Figure 5 sensors-23-06621-f005:**
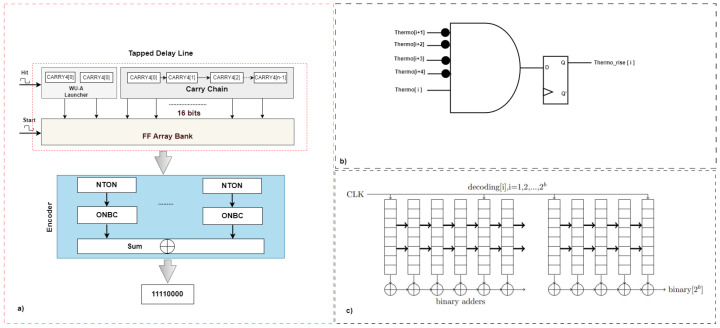
Encoder process. (**a**) Block diagrams. (**b**) Non-thermometer code to one-out-of-N code converter (NTON). (**c**) The one-out-of-N code to binary code converter (ONBC).

**Figure 6 sensors-23-06621-f006:**
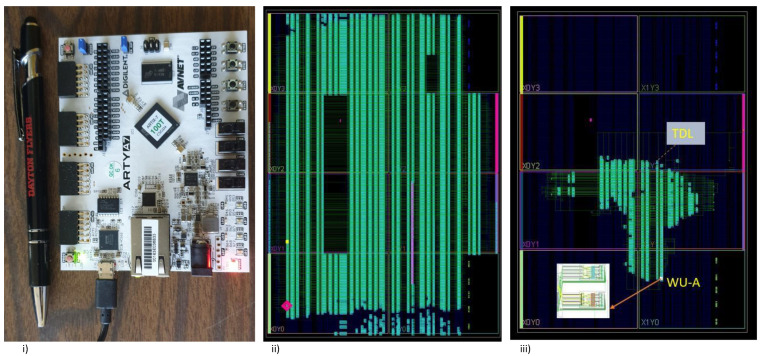
(**i**) Xilinx Arty-7 board. (**ii**) Floor plan of 32-channel WU-A TDC. (**iii**) Single-channel WU-A TDC implementation clock regions (X1Y1 X1Y0).

**Figure 7 sensors-23-06621-f007:**
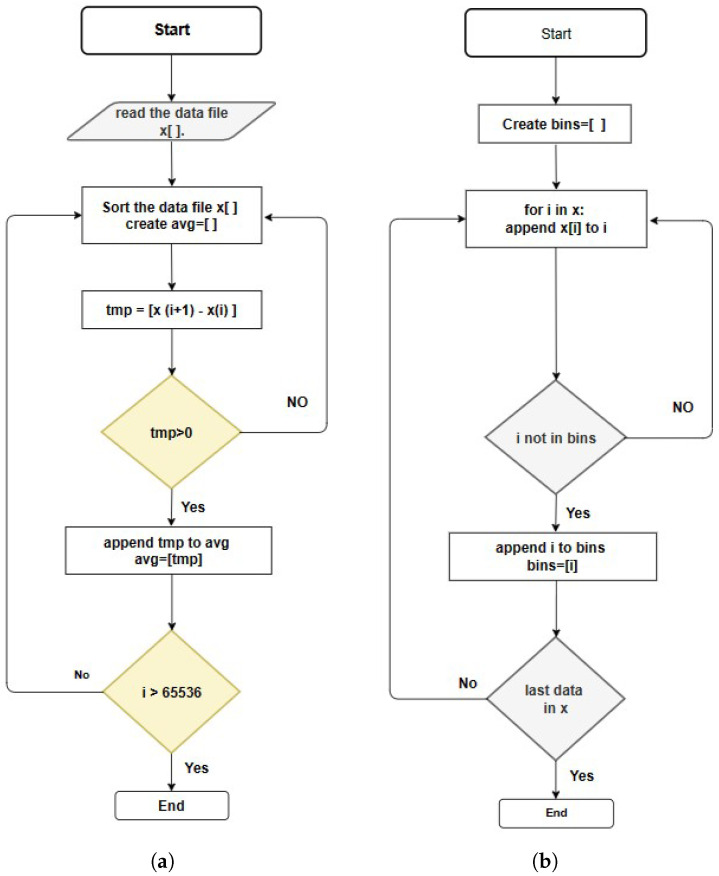
Algorithmic flowcharts for (**a**) the averaging process, (**b**) finding the active bins.

**Figure 8 sensors-23-06621-f008:**
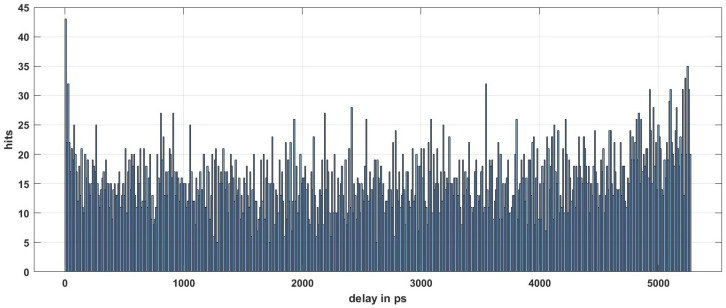
The graph shows the number of hits versus bins number.

**Figure 9 sensors-23-06621-f009:**
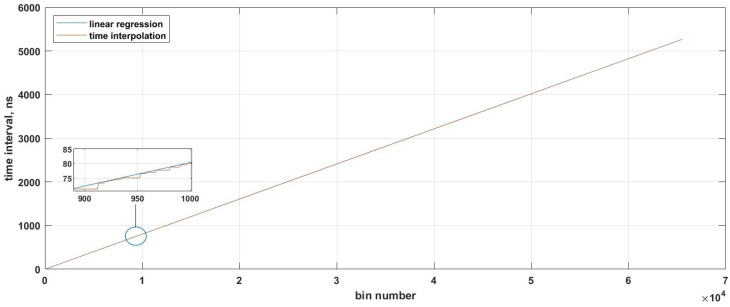
The graph shows the time interpolation linearity.

**Figure 10 sensors-23-06621-f010:**
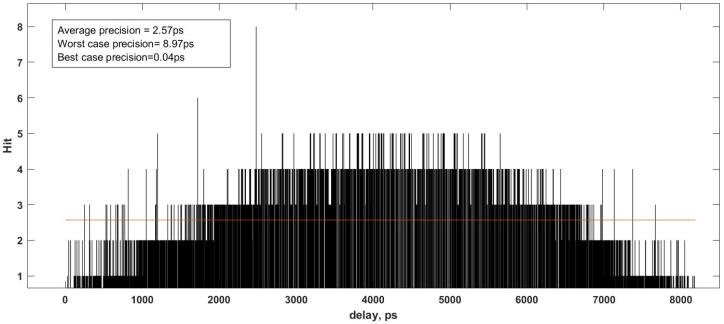
The graph shows the delay time versus the bin number, and the red horizontal line represents the average precision value.

**Figure 11 sensors-23-06621-f011:**
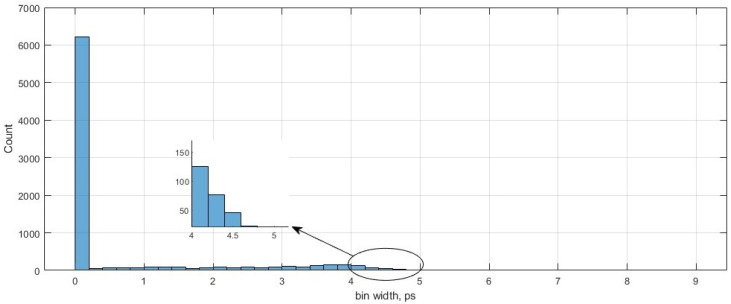
The graph shows the bin width distribution.

**Figure 12 sensors-23-06621-f012:**
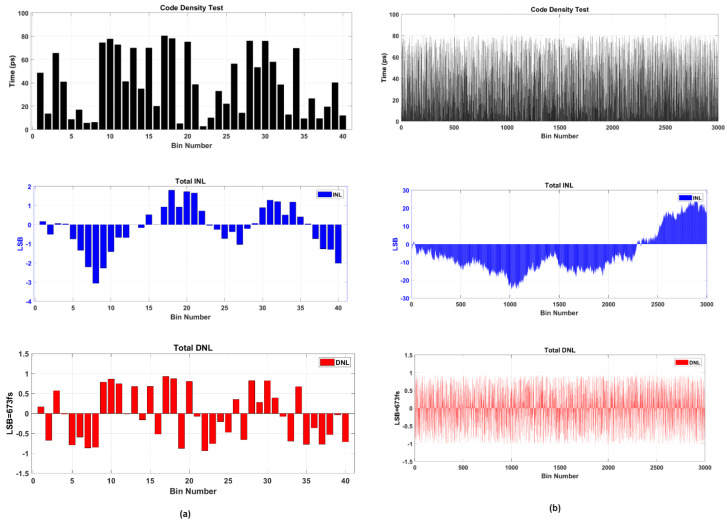
The graph shows code density and the linearities (DNL and INL) for the 32channels WU type A TDC (**a**) bin sizes tested at START bins, (**b**) bin sizes tested at 3000 bins.

**Figure 13 sensors-23-06621-f013:**
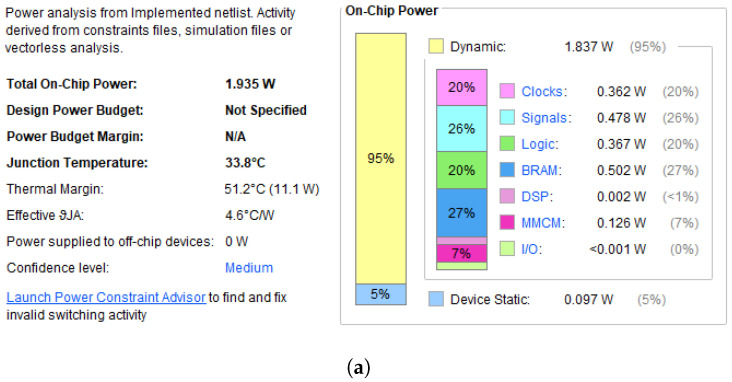
(**a**) Power analysis on-chip. (**b**) With the on-chip temperature around 66.5 °C.

**Table 2 sensors-23-06621-t002:** Survey of FPGA-based TDCs using Wave Union-A architecture.

Ref.	Board	Process	Method	RMS Resolution [ps]	DNL${}_{\mathrm{px}-\mathrm{pk}}$ (LSB)	INL${}_{\mathrm{px}-\mathrm{pk}}$ (LSB)
[23]	Cyclone II	90 nm	WU-TDL	21.00	N/A	N/A
[10]	Kintex-7	28 nm	TDL-TDC, WU-A Flip-Flop	10	N/A	N/A
[4]	Spartan-6	45 nm	Multichain 2-stage, WU-A LUT	6	[−1.00, 6.25]	[−26.2, 11.5]
[19]	Kintex-7	28 nm	WU-A-based (Multiplexers, TDL) (Eight edges)	3.90	[−1.00, 4.50]	[−37.70, 12]
[31]	Artix-7	28 nm	16 Ch-Super WU-TDL-TDC	12.5	N/A	[−3.7, 4.8]
[24]	UltraScale	20 nm	Sub-TDL, WU-A Compensation	3.58	[−0.92, 1.75]	[−1.20, 5.97]
[32]	Kintex-7	28 nm	Multisampling WU-A (4×transistions, 2×snapshots)	4.65	[−0.96, 4.26]	[−25.09, 4.26]
This Work	Artix-7	45 nm	32-channel, TDL-TDC, WU-A (Start in First Element)	1.81	[−1.01, +0.92]	[−25.24, +26.24]

**Table 3 sensors-23-06621-t003:** Specifications for FPGA Implementation Boards.

FPGA Board	Kintex-7	Virtex-5	Spartan-6	Zynq-7020	Artix-7	Ultra-Scale+
Process	28 nm	65 nm	45 nm	28 nm	45 nm	16 nm
Logic Cells	356,160	330,000	43,661	85,000	33,280	154,000
DSP Slices	1440	64	58	220	90	360
Memory	25,740	4608	2088	4900	1800	7600
I/O Pins	500	680	348	200	250	252
Clock Regions	12	16	12	8	8	12
CLBs	27,825	17,280	5831	34,675	7925	10,980

**Table 4 sensors-23-06621-t004:** On-chip resources report utilization.

Resource	WU-A 32-Channels Utilization (%)	Artix-7 Available
LUT	73.86%	63,400
LUTRAM	32.05%	19,000
FF	33.41%	126,800
BRAM	84.81%	135
Clock Regen	85% X0-Y0 to X1-Y3	8 Clocks regions

**Table 5 sensors-23-06621-t005:** The measurement of the proposed TDC-based WU-A method.

Parameter	Value/Range
Average Precision (ps)	2.57
Worst Case Precision (ps)	8.97
Best Case Precision (ps)	0.04
RMS Resolution (ps)	1.817
DNL, DNL${}_{pk\_pk}$(LSB)	1.923, [−1.01, +0.923] LSB
INL, DNL${}_{pk\_pk}$(LSB)	51.68, [−25.24, +26.24] LSB

## Data Availability

The data presented in this study are available upon request from the S.M.A. author.

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
