# Peer review of "A Size, Weight, Power, and Cost-Efficient 32-Channel Time to Digital Converter Using a Novel Wave Union Method"

_sensors, 2023, doi:10.3390/s23146621_

Round 1

Reviewer 1 Report

The paper describes the architecture of TDC and its implementation on an Xilinx FPGA.

The paper is composed of six parts, including an state of the art of existing TDC technologies, a description of the proposed architecture, its calibration method and its implementation and the results. I would like to suggest the author to add new discussion section before the conclusion (or to enhance the conclusion to include a discussion) in which you clearly state what is new in your approach and what is beyond the state of the art. This would highly improve the relevance of the paper and convince the reader about the originality of the approach.

I would also suggest the author to rewrite the description of the method and the calibration algorithm which is sometimes a bit messy. I suggest to pay attention to always use the term to describe the same concept. For instance, at the end of page 4, I guess that the number of sample corresponds to the number of bins? If yes, I suggest to use "number of bins" instead.

The convention used for formula and mathematical term is not usual. For instance, on line 128, [Wk] describes an vector and Wk the k-th term of the array. The common way to describe a vector is to use W (bold, non-italic) for the vector and W[k] (italic) or W_k (italic, k in index) to describe the terms of the vector.

I also have a concern about que quality and the relevance of the figures. Figure 2 has to be improved (very pixellized when zooming and zooming is required to understand the architectures). Figure 3 is a screenshot of the floorplan of the FPGA implementation, but the information of this figure is not exploited in the text, which makes the relevance of this figure questionnable.  I suggest to skip it. Same for 6b and 6c.

Author Response

Thank you for your valuable comments and your time and effort in reviewing the manuscript. I have attached our response to your comment. 

Reviewer 2 Report

Please avoid the use of abbreviations in abstract. Check and correct Fig. 11.

Please avoid the use of abbreviations in abstract. Check and correct Fig. 11.

Author Response

(The authors gave the same response as above.)

Reviewer 3 Report

In the paper by Saleh Alshahry, 

Awwad Alshehry, Abdullah K. Alhazmi and Vamsy P. Chodavarapu entitled “A Size, Weight, Power, and Cost-Efficient 32-Channel Time to Digital Converter using a Novel Wave Union Method” the. authors described the implementation of a multi-channel TDC using a SWaP-C efficient FPGA and demonstrated high-precision timing measurement. It included a novel WU-A method by connecting only the first multiplexers as input, and the output of this element is connected to the next multiplexers cascaded, repeatedly. As a result, it achieved high precision with using an efficient WU-A algorithm and calibration method. The results show that the proposed design provides high accuracy with an average time measurement precision of 2.57 ps. Furthermore they described an auto-calibration algorithm to help improve the TDC performance by improving the TDC DNL and INL.

The paper is organized as follows, Section II explains the architecture of the proposed 8WU-A launcher, and Section III describes the calibration algorithms. Section IV and Section V provide implementation details and results, respectively. Finally, Section VI concludes the work.

Figure captions are as follows:

Figure 1. Comparison of proposed and standard WU-A architecture.

Figure 2. i) Schematic diagram of complete TDC structure ii) Proposed WU-A architecture iii) Diagram of CARRY4 block of Xilinx Artix-7 FPGA [35].

Figure 3. Implemention layouts of the WU-A TDL TDC. i) Overview. ii) Clock regions (X1Y0 X1Y1) containing TDC CH00. iii) A single CLB for WU-A TDL TDC implementation.

Figure 4. Automatic calibration functional block.

Figure 5. Encoder process.(a) Block diagrams. (b) non-thermometer code to one-out-of-N code converter (NTON).(c) the one-out-of-N code to binary code converter (ONBC).

Figure 6. (a) Xilinx Arty-7 board (b) Floor plan of 32-channel WU-A TDC, (c) Single channel WU-A TDC implementation clock regions (X1Y1 X1Y0).

Figure 7. Algorithmic flowcharts for, (a) the averaging process, (b) finding the active bins.

Figure 8. The graph shows the number of hits versus bins number.

Figure 9. The graph shows the time interpolation linearity.

Figure 10. The graph shows the delay time versus bin number.

Figure 11. The graph shows the bin width distribution.

Figure 12. The graph shows code density and non-linearities (DNL and INL )for the 32-WU-A TDC (a) bin sizes tested at START bins. (b) bin sizes tested at 3000 bins.

Figure 13. (a) Power analysis on-chip. (b) With the on-chip temperature around 66.5°C.

A useful advance in the related area. I think the paper can be accepted as is.

Author Response

Thank you for your valuable comments and time and effort in reviewing the manuscript. I have attached our response to your comment. 

Reviewer 4 Report

In this manuscript, the authors present a Tapped Delay Line (TDL) based 32-channel Time to Digital Converter (TDC) with high precision. An auto-calibration algorithm is proposed to improve the Differential Non-Linearity (DNL) and Integral Non-Linearity (INL). The proposed TDC is implemented on a commercial Field-Programmable Gate Array (FPGA), Xilinx Artix-7, where a novel Finite Step Response Wave Union (WU-A) architecture of cascaded multiplexers is built to solve bubbles errors. By analyzing the test results of the TDC, it reveals an average time precision of 2.57 ps and an RMS precision of 1.817 ps. While this work can be contributing to high-precision time interval measurements with low size, weight, power and cost, there is still some questions about the manuscript might need to be addressed.

1) Are the terms “precision” and “resolution” used properly? I noticed that the authors claim 1.81 as RMS precision in Line 6 and 7, but as RMS resolution in Table 2 and 5. Clarification might be needed while the meaning of the two terms can be different.

2) By what benchmark the does the presented work compared with the state-of-the-art TDCs? While the proposed TDC is supposed to meet the SWaP-C requirements, advantages other than timing measurement might need to be demonstrated, especially when comparing with the works with different architectures and numbers of channels.

3) How does the proposed TDC perform in long-term? Will the cascaded WU-A architecture and high utilization of BRAM impact the long-term stability?

Author Response

(The authors gave the same response as above.)
